# L-arginine and Its Derivatives Correlate with Exercise Capacity in Patients with Advanced Heart Failure

**DOI:** 10.3390/biom13030423

**Published:** 2023-02-23

**Authors:** Anna Drohomirecka, Joanna Waś, Natalia Wiligórska, Tomasz M. Rywik, Krzysztof Komuda, Dorota Sokołowska, Anna Lutyńska, Tomasz Zieliński

**Affiliations:** 1Department of Heart Failure and Transplantation, National Institute of Cardiology, Alpejska 42, 04-628 Warsaw, Poland; 2Department of Medical Biology, National Institute of Cardiology, 04-628 Warsaw, Poland

**Keywords:** arginine, heart failure, ADMA, SDMA, nitric oxide, endothelium function, exercise capacity

## Abstract

Methylated arginine metabolites interrupt nitric oxide synthesis, which can result in endothelium dysfunction and inadequate vasodilation. Since little is known about the dynamics of arginine derivatives in patients with heart failure (HF) during physical exercise, we aimed to determine this as well as its impact on the patient outcomes. Fifty-one patients with HF (left ventricle ejection fraction-LVEF ≤ 35%, mean 21.7 ± 5.4%) underwent the cardiopulmonary exercise test (CPET). Plasma concentrations of L-arginine, citrulline, ornithine, asymmetric dimethylarginine (ADMA), and symmetric dimethylarginine (SDMA) were measured before and directly after CPET. All patients were followed for a mean of 23.5 ± 12.6 months. The combined endpoint was: any death, urgent heart transplantation, or urgent LVAD implantation. L-arginine concentrations increased significantly after CPET (*p* = 0.02), when ADMA (*p* = 0.01) and SDMA (*p* = 0.0005) decreased. The parameters of better exercise capacity were positively correlated with post-CPET concentration of L-arginine and inversely with post-CPET changes in ADMA, SDMA, and baseline and post-CPET SDMA concentrations. Baseline and post-CPET SDMA concentrations increased the risk of endpoint occurrence (HR 1.02, 95% CI 1.009–1.03, *p* = 0.04 and HR 1.02, 95% CI 1.01–1.03, *p* = 0.02, respectively). In conclusion, in patients with HF, extensive exercise is accompanied by changes in arginine derivatives that can reflect endothelium function. These observations may contribute to the explanation of the pathophysiology of exercise intolerance in HF.

## 1. Introduction

Heart failure (HF) is a multifactor-conditioned clinical syndrome that affects all systems and organs of the human body. The pathophysiology of HF includes alterations in the hemodynamics, but also a broad spectrum of neurohormonal disorders. Unfortunately, many potential mechanisms are still poorly clarified. Moreover, some phenomena observed in in vitro studies are not as easy to translate in clinical conditions and to be defined by simple diagnostic methods. Endothelium dysfunction plays an important role in the development and progression of HF and has been thoroughly investigated. Except from many different actions, the endothelium is responsible for the vascular tone regulation via production of vasoconstrictors and vasodilators.

Nitric oxide (NO), formed in the endothelium from L-arginine and oxygen by the enzyme NO synthase (NOS), is one of the main factors involved in the redistribution of regional blood flow through smooth muscle dilation and myofibrillar relaxation [1]. Therefore, NO is one of the crucial players in vascular tone regulation, and impaired NO synthesis indicates deterioration of normal endothelium activity. The metabolism of arginine, a substrate for NO synthesis, at the cellular level is well described [2,3]. Briefly, arginine is oxidized by NOS to citrulline and NO. It can be expected that NO synthesis decreases when the availability of arginine is limited or the activity of NOS is diminished or blocked. NO synthesis may be interrupted by methylated arginines. Asymmetric dimethylarginine (ADMA) is a direct inhibitor of NOS that competes with arginine for NOS binding. Furthermore, both ADMA and symmetric dimethylarginine (SDMA) interfere with NO synthesis indirectly by reducing the availability of the cationic amino acid transporters (CAT) for arginine, as they all use the same CAT for the transport across the cell membrane [4]. Arginine takes also part in urea cycle, where it is converted by arginase in ornithine and urea. Thereby, arginase, by competing for arginine with NOS, becomes a modulator of NO production.

The Imbalance in the NO pathway implies worse clinical outcomes. Numerous studies have highlighted the relationship of dimethylarginines with cardiovascular complications. Elevated levels of ADMA and SDMA were found to be risk markers for total and cardiovascular mortality in a meta-analysis of prospective studies in different populations [5], were associated with mortality in acute heart failure patients [6] and were predictors of major adverse cardiovascular events in patients undergoing elective diagnostic cardiac catheterization [7]. In patients with HF, ADMA and SDMA were higher in individuals with more advanced symptoms (higher NYHA class) [8] and correlated with the well-known HF biomarker, the natriuretic peptide NT-pro-BNP [9]. Interestingly, ADMA has also been shown to increase the risk of a new incidence of HF, independently of myocardial infarction appearance [10]. Therefore, disturbances in NO production and arginine metabolism may be the link between HF and endothelium dysfunction.

Although a key symptom of HF is an impaired exercise capacity, patient-adjusted exercise training is recommended to improve physical capacity, quality of life and prognosis (by reduction in HF related hospitalizations) [11]. In addition, a beneficial effect of exercise training on endothelium function, expressed as agonist-mediated endothelium-dependent vasodilation of the skeletal muscle vasculature, together with a significant increase in exercise capacity, was demonstrated in patients with HF [12]. Consistent with that, it seems obvious that exercise causes changes in arginine metabolism related to NO production. Furthermore, since blood redistribution during exercise is also, to some extent, controlled by endothelium-derived NO, reduced exercise capacity in HF may be partially explained by the impaired exercise-induced hyperemic blood flow response resulting from endothelium dysfunction in HF [1]. However, data on changes in arginine derivatives in response to exercise are limited mostly to patients with coronary artery disease and healthy subjects, and differ depending on subject health status or type of exercise (regular long-term training, one-off intensive physical activity). To the best of our knowledge, there is no evidence regarding variations in plasma concentrations of arginine derivatives during exercise in patients with HF.

Therefore, we hypothesized that extensive exercise in patients with HF provokes changes in concentration of L-arginine derivatives which could be measured in plasma and the pattern of those changes may differ dependent on HF progression. Consequently, our objective was to determine the pattern of changes in L-arginine derivatives after a submaximal exercise test in patients with HF and its correlation with exercise capacity and patients’ prognosis.

## 2. Materials and Methods

### 2.1. Study Population and Design

This is a prospective study conducted as a statutory grant (No. 2.47/VII/20) at a tertiary cardiology center.

The study population comprised 51 patients, 36–66 years old (mean 55.6 ± 7.5), *n* = 8, 16% women, with chronic heart failure admitted to our center for evaluation of heart transplant candidacy. Patients presented with class II–III New York Heart Association (NYHA) function and left ventricular ejection fraction (LVEF) <35% (mean 21.7 ± 5.4%) as evaluated by echocardiography. The detailed population characteristics are presented in Table 1. Exclusion criteria were: administration of catecholamines, contraindications to performance of cardiopulmonary exercise test (CPET), pneumonia or bronchitis within last two weeks, or severe ventilation disorders with forced expiratory volume in one second (FEV1) < 50%.

All patients were pharmacologically treated according to the European Society of Cardiology guidelines in force at the time of enrollment.

In all subjects, aerobic capacity was measured with a CPET on a bicycle ergometer (product of Lode Medical Technology). Calibration of the system was performed before each test. The exercise test consisted of three minutes of unloaded pedaling followed by graded workload increase of 10 Watt every minute. All tests were terminated on patients’ request due to their symptoms (extensive dyspnea or fatigue). The expiratory exchange ratio (RER) was defined as carbon dioxide output (VCO_2_) divided by oxygen consumption (VO2). The relation between minute ventilation and carbon dioxide production was expressed as VE/VCO_2_ slope. Peak VO_2_ was defined as the highest 30-s average value obtained during the period directly before exercise termination. Peak VO_2_ adjusted for age and sex was calculated automatically by the system software. After terminating the exercise, patients spent at least 2 min in a cool-down period on the bicycle with no workload. The results of CPET are summarized in Table 1.

The patients were followed for a mean of 23.5 ± 12.6 months and the date of death, heart transplantation, or left ventricular assist device (LVAD) implantation was considered as the end of the observation. Death, urgent heart transplantation or urgent LVAD implantation were defined as the composite end point.

The study complied with the Declaration of Helsinki and was approved by the regional Ethics Committee (No. of approval 1853). All patients gave their written consent to participate in the study.

### 2.2. Blood Sampling and Laboratory Methods

Venous blood samples were drawn at rest and directly after the completion of the CPET (after a rest period of 10 to 15 min). Plasma was frozen immediately after centrifugation and stored at −80 °C until analysis. Plasma concentrations of L-arginine, ADMA, SDMA, ornithine, and citrulline were measured by liquid chromatography tandem mass spectrometry (LC–MS/MS).

Since the arginine/ADMA ratio reflects the proportion of two competing substances (arginine as a substrate for NOS and ADMA as its inhibitor), we used it as an indicator of NO production. Moreover, the global arginine bioavailability ratio (GABR = arginine/(ornithine + citrulline)) was calculated.

#### 2.2.1. Chemicals

The certified analytical kit AccQ-Tag Ultra Derivitization Kit dedicated for Amino Acid Analysis (AAA) was purchased from Waters (Milford, MA, USA).

Ornithine (ORN), citrulline (CIT), and arginine (ARG) were purchased from Cambridge Isotope Laboratories (Tewksbury, MA, USA), and ADMA, SDMA, and isotope internal standards (IS) were purchased from Toronto Research Chemicals (Toronto, ON, Canada).

#### 2.2.2. Instrumentation

Arginine derivative measurements were performed using an enhanced high-performance hybrid triple quadrupole/linear ion trap LC-MS/MS mass spectrometer-QTRAP 6500+ (Sciex Framingham, MA, USA). The UHPLC system includes the following EXION LC components: an autosampler, a binary solvent pump, a degasser, and a column oven. The universal laboratory centrifuge (5417C, Eppendorf, Hamburg, Germany) was used for sample preparation. Data analysis was obtained with Analyst software v.1.7 for acquisition and processing. Multi Quant 3.03. was used for quantification of the results.

#### 2.2.3. Mass Spectrometry Conditions

The separation of compounds was performed using a Phenomenex Luna^®^ Omega 3 µm PS C18 column (2.1 × 100 mm) maintained at 40 °C. A binary step–gradient at a flow rate of 0.45 mL/min was used. The mobile phases were a mixture of Phase A: water-ammonium formate (10 mM) and Phase B: methanol-ammonium formate (10 mM). The gradient program was initially set at 85% (A) followed by a switch to 95% (B) at 10.4 min, and after a switch back to 85% (A) at 13.0 min. The total chromatographic analysis time was 13 min. The mass spectrometer operated in multiple reaction monitoring (MRM) scanning mode. The Turbo Spray Ion Drive source was used for application in positive electrospray ionisation ESI. One parent ion was selected in Q1, daughter ions were detected in Q3. The selective MRM transitions (*m*/*z*) were as follows: ORN- [303.2 → 171.2] ORN-IS [308.2 → 171.1]; ARG- [345.2 → 175.1], ARG-IS [355.2 → 171.1]; CIT- [346.2 → 171.2], CIT-IS [351.2 → 171.1]; ADMA [373.2 → 203.2], ADMA-IS [379.2 → 209.2]; SDMA [373.2 → 171.2], SDMA-IS [379.2 → 175.2].

#### 2.2.4. Calibration

The precipitation reagent contained a mixture of the isotope internal standard (1 mg/mL) in methanol-water (20:80 *v*/*v*). The derivatization reagents (AccQ-Tag Ultra Reagent Powder 2A) were dissolved in 1 mL of acetonitryle (AccQ-Tag Ultra Reagent Diluent 2B). Subsequently, the vial was placed on a heating block (55 °C) for 10–15 min. The stock solution of the analytes was prepared in a water AT concentration of (1 mg/mL). The concentrations of the 6 individual calibrations were 10, 50, 100, 250, 500, and 1000 ng/mL for ADMA and SDMA; 9 individual calibrations 10, 50, 100, 250, 500, 1000, 2500, 5000, and 10,000 ng/mL for ARG and ORN; and 13.3, 66, 131, 328, 657, 1314, 3285, 6570, and 13,300 ng/mL for CIT.

The curves were calculated using a weighted linear regression analysis with w = 1/x implemented to improve the adjustment at low concentrations with consistent correlation coefficients of 0.99.

#### 2.2.5. Sample Preparation

A volume of 50 µL of serum was transferred to a 1.5 mL standard Eppendorf tube, then mixed with 200 µL of protein precipitation reagent, which was a mixture of internal standards in water-methanol (20:80 *v*/*v*). The samples were vortexed for 15 min and centrifuged (3000 RPM, 10 min). Next, 10 µL of supernatant was transferred to a new vial with insert, then 75 µL of borate buffer (Acc-Tag Borate Buffer) and 15µL of reagent 2A were added. The resulting mixture was mixed (5 s, 450 RPM) and incubated for 1 min at room temperature. Then it was placed into a heating block (55 °C) for 10 min. Subsequently, 900 µL of water was added and vortexed for 1 min. Finally, it was put to an autosampler and 10 μL was injected into the UPLC-MS/MS.

### 2.3. Statistical Methods

Continuous data were presented as mean ± SD, dichotomous variables were presented as proportion. The Shapiro–Wilk test was used to verify whether a variable is normally distributed. Differences in values of repeated measurements within study population were measured using the paired samples-t test or Wilcoxon test when appropriate. Differences between groups (e.g., patients receiving vs. non-receiving sildenafil) were tested with two sample t-test or Mann–Whitney U test when appropriate. The relation between continuous variables was evaluated using the Spearman correlation test. Univariable Cox regression analysis was performed to identify predictors of event free survival.

Statistical analysis was performed using Statistica 12.0 software. A *p*-value of less than 0.05 was considered statistically significant.

## 3. Results

### 3.1. Concentrations of L-arginine Derivatives before and after CPET

Except citrulline, all arginine derivatives analyzed showed changes in concentrations after CPET compared to baseline values: ornithine, SDMA and ADMA concentrations decreased while arginine increased. Consequently, the ARG/ADMA ratio and GABR were higher after CPET. Details are presented in Table 2.

### 3.2. Association between L-arginine Derivatives and Exercise Capacity

We analyzed the correlation between all measured L-arginine derivatives before CPET and after CPET, as well as the absolute differences (value after CPET minus value before CPET) and the parameter of exercise capacity (pVO_2_ adjusted for age and sex, VE/VCO_2_ slope, RER). Sex and age adjusted peak VO_2_ was positively correlated with post-CPET concentration of arginine, citrulline, post-CPET ARG/ADMA ratio, post-CPET vs. baseline changes in citrullin, and inversely with baseline and post-CPET SDMA. VE/VCO_2_ slope correlated with post-CPET ornithine concentration, post-CPET vs. baseline changes in ADMA and SDMA. The respiratory exchange ratio (RER) was inversely correlated with post-CPET concentration of arginine, post-CPET ARG/ADMA ratio and positively with post-CPET SDMA concentration (both as an absolute value and as an increase vs. baseline). Table 3 presents detailed data with correlation coefficients and *p* values restricted to parameters which were statistically significantly correlated.

### 3.3. L-arginine Derivatives Concentrations Depending on Patient Medication

As beta-blockers, angiotensin-converting enzyme inhibitors/angiotensin receptor blockers/angiotensin receptor–neprylisin inhibitor were administered in all patients and spironolactone/eplerenone in all but one patient; the impact of those drugs on arginine metabolism were not analyzed.

In patients receiving sildenafil, baseline and post-exercise concentrations of ornithine were lower than in patients without sildenafil treatment (*p* = 0.0.02 and *p*-0.04, respectively). A significant increase in post-exercise arginine concentration (vs. baseline values) was observed in patients who were treated with statins compared with patients who were not (*p* = 0.030).

### 3.4. L-arginine Derivatives and Patient Prognosis

Patients who met the criteria of composite end point in comparison to event-free survivors had higher concentrations of SDMA after CPET (70.7 ± 21.3 vs. 60.4 ± 16.8 ng/mL, *p* = 0.04) tendency to higher concentrations of SDMA before CPET (74.2 ± 21.8 vs. 63.1 ± 19.3 ng/mL, *p* = 0.054) and lower exercise capacity (pVO_2_ 9.6 ± 2.3 vs. 11.2 ± 3.2 ml/kg/min, *p* = 0.07). The remaining L-arginine derivatives concentrations and exercise capacity parameters (pVO_2_ adjusted for sex and age, VE/VCO_2_ slope, RER) did not differ between the two groups.

Both baseline and post-CPET concentrations of SDMA increased the risk of composite endpoint occurrence (HR 1.02, 95% CI 1.009–1.03, *p* = 0.04 and HR 1.02, 95% CI 1.01–1.03, *p* = 0.02, respectively). None of the other L-arginine derivatives, neither any of exercise capacity parameters had any impact on patients’ prognosis.

## 4. Discussion

In our study, we focused on patients with HF undergoing a symptom-limited submaximal exercise test. We chose a model of forced exercise assuming that changes in arginine derivatives would have been mostly expressed during maximal intensity exercise exertion. After CPET, we observed a significant increase in plasma arginine concentrations, whereas the concentrations of ADMA and SDMA decreased. The results are consistent with those presented by Riccioni et al. [13] in the population of patients with coronary artery disease (CAD); however, their exercise protocol was little different consisting of combination of aerobic conditioning (walking on a treadmill), muscle strengthening, and increasing joint flexibility activities. In addition, we also managed to show that the indicators of arginine availability (arginine/ADMA ratio, GABR) improved after submaximal exercise. The arginine/ADMA ratio was already postulated as a marker of the balance between L-arginine and ADMA competing in the activation of endothelial NOS [14] and its relationship with better survival was proven in patients with dilated cardiomyopathy [8]. GABR, on the other hand, represents the overall balance between arginine and arginine catabolism by arginase and NOS, and its lower values were demonstrated to be predictors of more advanced HF and worse prognosis [15].

Based on our research, there is no direct proof that changes in the amounts of arginine derivatives observed in the circulating blood reflect their amount in endothelial cells and other tissues. We considered the hypothesis that an increase in plasma arginine concentration may be a consequence of decreased myocardial and systemic arginine uptake, which was previously described in patients with HF [16]. However, a higher arginine concentration and the arginine/ADMA ratio after CPET were correlated with better exercise test results, while an inverse correlation was observed with an increase in SDMA and ADMA after CPET. It suggests that an increase in plasma arginine with a concomitant decrease in ADMA and SDMA enhanced exercise capacity rather than being a random or harmful phenomenon. It is consistent with the observation that the production of nitric oxide depends on extracellular L-arginine [17].

However, the mechanism of arginine derivatives’ homeostasis is much more complicated and cannot be simply explained in the light of contrasting results of other researchers. To the best of our knowledge, our study is the first one to determine arginine and its four derivatives before and after submaximal exercise in patients with HF. Ilic et al. [18] reported that after exercise stress echocardiography plasma levels of ADMA and SDMA increased both in patients with CAD and in healthy volunteers, with a higher baseline and post-exercise value in the CAD group. In contrast to that, the results of Pawlak-Chaouch et al. [19] did not demonstrate any change in SDMA and ADMA after a maximal graded exercise test in healthy men independently of their physical capacity. Plasma ADMA and SDMA were not associated with the pattern of local muscle deoxygenation and exercise capacity. Unlike in our study group, in healthy men examined by Pawlak-Chaouch et al. [19], plasma arginine concentrations were significantly lower after exercise compared to baseline. Moreover, the arginine/ADMA ratio (both baseline and after exercise values) was inversely correlated with exercise capacity (expressed as maximum oxygen uptake) in their study [19], whereas we demonstrated positive correlation between the arginine/ADMA ratio after CPET with peak oxygen consumption. Last, but not least, a decrease in arginine concentration accompanied with a decrease in arginine/ADMA ratio and increase in SDMA was observed in participants of Norseman Xtreme triathlon (an ironman distance triathlon) after the completion of the race [20].

Different patterns of changes in plasma concentrations of arginine derivatives in the response to exercise in HF patients and healthy individuals may result from chronic alternations of arginine transport and metabolism, as well as compensatory mechanisms in HF. An in vitro study has shown that hypoxia decreased CAT-2 expression, L-arginine uptake, and NO production in human pulmonary microvascular endothelial cells. However, overexpression of CAT-1 attenuated the hypoxia-induced decrease in NO production [21]. On the contrary, a significant decrease in CAT-1 mRNA expression was observed in ventricular myocardial samples from patients with HF compared with healthy unused donor myocardium, while myocardial NOS enzymatic activity and NOS protein were unchanged [16]. The potential role of CAT and its inhibition by methylated arginines in endothelium dysfunction may explain the results of our study. We have demonstrated that increased SDMA was associated with worse exercise capacity and, as the only arginine derivatives, with poor prognosis. Until now, SDMA was given less attention than ADMA due to a lack of direct NOS inhibition capacity. Since SDMA is mainly cleared by the kidneys, it was initially considered a marker of renal function. The latest study of Lorin et al. [22] reported, however, that SDMA predicts HF development in patients hospitalized for acute myocardial infarction independently of estimated glomerular filtration rate with accuracy similar to that of NT-pro-BNP.

Endothelium function and NO synthesis may be influenced by pharmacotherapy. An impact of beta-blockers and agents affecting the rennin–angiotensin–aldosterone system should be homogenous in the study group, as all patients were treated with those groups of drugs. Sildenafil enhances the effect of NO through inhibiting cyclic guanosine monophosphate (cGMP) hydrolysis [23]. In our study, we observed that patients treated with sildenafil had lower concentrations of ornithine. Ornithine is a product of arginine degradation by arginase. In an experimental study, Tai et al. [24] showed that sildenafil downregulated arginase 1. This type of action could explain the results we obtained. Another group of medicines with a proven positive effect on endothelium are statins [25]. As we presented, in patients receiving statins, an increase in arginine concentrations after exercise test were greater when compared with those without statin therapy. Previously, Tun et al. [26] showed that simvastatin treatment elevated L-arginine uptake, the expression levels of CAT-1 and eNOS mRNA, and NO production. The aforementioned observations could suggest that treatment with statins not only increases the amount of L-arginine but also improves its bioavailability.

### Limitation of the Study

We enrolled a very selected group of patients with HF–potential candidates for heart transplantation. This means that the results may not be simple generalized. Patients considered as heart transplant candidates are usually younger and have less comorbidity than an average patient with HF. Although the sample size is not large, in comparison to other studies evaluating baseline and post-exercise arginine derivatives, it is one of the biggest.

Moreover, we did not directly assess the concentration of NO nor NOS activity.

None of the patients received supplementation of L-arginine or L-citrulline. However, we did not analyze in details patient diet in respect of natural components which could change arginine metabolism.

The study population comprised only patients with heart failure without healthy subject control group. That is why we discussed our results only in the context of other researchers results.

## 5. Conclusions

In conclusion, the results of the present study demonstrated that in patients with HF, extensive exercise is accompanied by changes in arginine derivatives that may reflect NO availability and endothelium function. These observations may contribute to the explanation of the pathophysiology of exercise intolerance in HF. As for now, the results have mostly scientific value but, based on them, there is a justified need for further research which could define clinical interventions with potential benefits in patients with HF.

## Figures and Tables

**Table 1 biomolecules-13-00423-t001:** Clinical characteristics of the study population. ACEI = angiotensin-converting enzyme inhibitor, ARB = angiotensin receptor blocker, ARNI = angiotensin receptor–neprylisin inhibitor, COPD = chronic obstructive pulmonary disease, CRT-D = cardiac resynchronization therapy defibrillator, eGFR = estimated glomerular filtration rate, HF = heart failure, ICD = implantable cardioverter defibrillator, LVEDD = left ventricular end-diastolic dimension, LVEF = left ventricular ejection fraction, NYHA = New York Heart Association, MRA = mineralocorticoid receptor antagonist, NT-pro-BNP = N-terminal prohormone of brain natriuretic peptide, pVO2 = peak oxygen uptake, RER = respiratory exchange ratio, TAPSE = tricuspid annular plane systolic excursion, VE/VCO_2_ = minute ventilation/carbon dioxide production * mitral or tricuspidal regurgitation at least moderate.

	Study Population (*n* = 51)
**Clinical Characteristics**	
Age, years	55.6 ± 7.5
Gender, % male	43 (84.3%)
Ischemic etiology of HF	31 (60.8%)
Atrial fibrillation	24 (47.1%)
Hypertension	17 (33.3%)
Diabetes mellitus	20 (39.2%)
COPD	5 (9.8%)
ICD/CRT-D	30 (58.8%)/14 (27.5%)
II NYHA class, patients (%) II/III NYHA class, patients (%) III NYHA class, patients (%)	12 (23.5%) 10 (19.7%) 29 (56.9%)
**Echocardiographic parameters**	
LVEF [%]	21.7 ± 5.4
LVEDD [mm]	73.5± 9.6
TAPSE [mm]	17.0 ± 3.3
Mitral regurgitation *, patients (%)	37 (72.5%)
Tricuspid regurgitation *, patients (%)	24 (47.1%)
**Laboratory test results**	
NT-pro-BNP [pg/mL ], median (interquartile range)	2604 [1185–4827]
eGFR, ml/min/1.73 m^2^	59.3 ± 16.1
bilirubin [mmol/l]	19.4 ± 9.1
**Medication**	
Beta-blockers [*n*, %]	51 (100.0)
Target beta-blocker dose achieved [*n*, %]	19 (37.3)
ACEI/ARB/ARNI [*n*, %]	51 (100.0)
Spironolactone/eplerenone	50 (98.0)
Loop diuretics [*n*, %]	50 (98.0)
More than one diuretic administered (except from MRA) [*n*, %]	26 (51.0)
Sildenafil, patients [*n*, %]	11 (21.6)
Allopurinol, patients [n, %]	21 (41.2)
**Cardiopulmonary exercise test results**	
pVO_2_ [ml/kg/min]	10.5 ± 2.9
pVO_2_ adjusted for sex and age [%]	38.6 ± 12.6
RER at peak exhaustion	1.1 ± 0.1
VE/VCO_2_ slope	43.9 ± 12.3

**Table 2 biomolecules-13-00423-t002:** Changes in arginine derivatives after cardiopulmonary exercise test. ADMA = asymmetric dimethylarginine, CPET = cardiopulmonary exercise test, GABR = global arginine bioavailability ratio, SDMA = symmetric dimethylarginine.

	Basic Value (Mean ± SD)	After CPET (Mean ± SD)	*p* Value
arginine [μL/mL]	10.8 ± 3.6	11.4 ± 3.6	0.02
ADMA [ng/mL]	127.3 ± 20.8	122.4 ± 19.4	0.01
SDMA [ng/mL]	67.5 ± 20.8	64.4 ± 19.2	0.0005
ornithine [μL/mL]	6.7 ± 1.8	6.2 ± 1.6	0.002
citrulline [μL/mL]	6.0 ± 2.1	6.1 ± 2.2	0.37
arginine/ADMA ratio	86.7 ± 30.4	94.8 ± 30.3	0.006
GABR	0.87 ± 0.27	0.94 ± 0.27	0.003

**Table 3 biomolecules-13-00423-t003:** Correlation between the concentration of arginine derivatives and parameters of exercise capacity measured by cardiopulmonary exercise test. Data are presented as correlation coefficient and *p* value; only statistically significant correlations are presented. ADMA = asymmetric dimethylarginine, ARG = arginine, CIT = citrulline, CPET = cardiopulmonary exercise test, NS = not statistically significant, ORN = ornithine, pVO2 = peak O_2_ consumption, R = correlation coefficient, RER = the expiratory exchange ratio, SDMA = symmetric dimethylarginine, VE/VCO_2_ slope = the relation between minute ventilation and carbon dioxide production.

	Sex and Age Adjusted pVO_2_	VE/VCO_2_	RER
ARG post-CPET	R = 0.30 *p* = 0.03	NS	R = −0.38 *p* = 0.005
CIT post-CPET	R = 0.30 *p* = 0.03	NS	NS
CIT post-CPET minus baseline value	R = 0.32 *p* = 0.02	NS	NS
ORN post-CPET minus baseline value	R = 0.3 *p* = 0.03	R = 0.37 *p* = 0.007	NS
ADMA post-CPET minus baseline value	NS	R = 0.36 *p* = 0.009	NS
SDMA pre-CPET	R = −0.28 *p* = 0.048	NS	R = 0.39 *p* = 0.004
SDMA post -CPET	R = −0.31 *p* = 0.025	NS	R = 0.38 *p* = 0.005
SDMA post-CPET minus baseline value	NS	R = 0.28 *p* = 0.046	NS
(post-CPET ARG)/(post-CPET ADMA) ratio	R = 0.27 *p* = 0.053	NS	R = −0.34 *p* = 0.015

## Data Availability

Data are available on request from Anna Drohomirecka adrohomirecka@tlen.pl.

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
