# Peer review of "L-arginine and Its Derivatives Correlate with Exercise Capacity in Patients with Advanced Heart Failure"

_biomolecules, 2023, doi:10.3390/biom13030423_

Round 1
Reviewer 1 Report
I read with interest the manuscript entitled "L-arginine and its derivatives correlate with exercise capacity in patients with advanced heart failure" by Anna Drohomirecka et Al.
The paper is well written and methodically well designed. The authors conclude, in accordance with literature, that the endothelium dusfunction plays an important role to explanation of the pathophysiology of exercise intolerance in HF patients.
In order to make the work more interesting for Biomolecules 's readers there are some further elements to add:
- in the "results" section, at page 6, report data about the concentration of arginine derivatives and parameters of exercise capacity measured by CPET in patients with events versus patients without events during follow-up. Moreover, share these results in the discussion section.
- at page 7, line 48, delete reference number 19 (there is not a proper correlation between what is described in the paper and the reference's topic).
Author Response
Dear Reviewer, thank you very much for your positive feedback and all your
comments, which helped us to improve our manuscript.
Hereby we present the answers for your suggestions:
- in the "results" section, at page 6, report data about the concentration of arginine derivatives and parameters of exercise capacity measured by CPET in patients with events versus patients without events during follow-up. Moreover, share these results in the discussion section.
We added the information about L-arginine derivatives concentrations and indicators of exercise capacity in patients who met the composite end point and event-free survivors.
- at page 7, line 48, delete reference number 19 (there is not a proper correlation between what is described in the paper and the reference's topic).
We were quite confused with your comment about one of references, therefore we carefully analyzed once again the paper of Pawlak-Chaouch et al. [19] to verify whether we choose the reference correctly. We cited the paper of Pawlak-Chaouch et al. [19] to present the background of physiologic changes of L-arginine derivatives in healthy subjects during exercise. In our study we did not include a control group that is why we supported the discussion with other researchers’ results. Especially, that the lack of control group was raised as a limitation of our study by one of the reviewers. As some sentences in the discussion could have been misleading, we rewrote this part. We hope that it clarifies the relation between our results and the paper of Pawlak-Chaouch et al. [19]. Otherwise we would highly appreciate further explanation because maybe we just did not understand you comment correctly.
Reviewer 2 Report
This manuscript was designed to determine the pattern of changes in L-arginine derivatives after a submaximal exercise test in patients with HF and its correlation with exercise capacity and patients’ prognosis. The manuscript is interesting, but some concerns needs to be addressed
Abstract
What does mean LVEF?
What does mean ADMA and SDMA?
Introduction
The authors did not provide a background with respect to the exercise.
What was the study´s hypothesis?
The introduction is not clear enough to understand the relationship between exercise and L-arginine derivates
Did the authors control the participant’s diet? For example, there is food presenting L-citrulline (watermelon, cucumber, etc.) which could influence the study’s findings.
Author Response
Dear Reviewer,
We would like to thank you for the time and effort taken to review our manuscript.
We appreciate all your comments and concerns which helped us to improve the paper.
In the revised version of the manuscript we followed your suggestions:
Abstract
What does mean LVEF?
What does mean ADMA and SDMA?
- we explained the abbreviation in the abstract
Introduction
The authors did not provide a background with respect to the exercise.
We rewrote the introduction
What was the study´s hypothesis?
In our study we hypothesized that after intensive exercise some changes of L-arginine derivatives would be observed in patients with heart failure and be related to patients outcomes (physical capacity and prognosis). We rewrote the part of the manuscript where the objective of the study is described.
The introduction is not clear enough to understand the relationship between exercise and L-arginine derivates
We rewrote the introduction
Did the authors control the participant’s diet? For example, there is food presenting L-citrulline (watermelon, cucumber, etc.) which could influence the study’s findings.
- we added some more information about limitation of the study (we did nor analyze in details patient diet, however none of the patients took a supplementation of arginine or citrulline)
Reviewer 3 Report
Thank you for the opportunity to review the manuscript again. First, however, I would like to note that the current version does not contain line numbers, so it is difficult to point out all the corrections. Please take this into account in the next revision.
The manuscript still addresses an interesting topic. Overall, the manuscript has been revised mainly in form. There are no significant improvements in content. In my view, the manuscript needs to be revised at further points. I, therefore, ask the authors to address the following major points:
Introduction:
Overall, the introduction is short and not very informative. Only a few references are used. In particular, the definition of the research question is not comprehensible.
The interaction of HF, NO pathway, L-arginine derivative, and exercise capacity is not clearly described.
It is described that "little is known about variations in plasma ..." but no further details are given. What is already known about this?
Therefore, the aim of the study is not fully clear. Why is it essential to evaluate the L-arginine derivative changes after submaximal exercise?
Methods:
The methods of the study are well-described, detailed, and comprehensible.
Please correct under point 2.3 "Statistical methods": Differences between Groups... Were two groups formed or do you mean "Within Group"?!
Results:
There are important points to be revised in this part of the paper:
I cannot understand table 3. What are the reasons for choosing these parameters? Why is once the POST value was taken, once the PRE value, and once the "difference"? I don't see a pattern here. What about the other values? Why are many values missing in the table? The selection seems a bit random.
The risk of endpoint occurrence - how was this measured? Please explain a little more about "statistical methods". Are there different endpoints measured? In parallel to the correlations in table 3: what about the results of other arginine derivatives?
In total, the second part of the results section is not well structured. Please revise.
Discussion:
The first half of the discussion refers to the basics of the introduction and does not include the actual results. Facts are mentioned here that are missing in the introduction. I, therefore, recommend that the introduction be supplemented with the contents of the discussion.
The discussion of the results is well done. About the results section, is it also possible to make statements about a norm collective?
Please also address a possible clinical implication in the discussion.
In your opinion, is the lack of a control group a limitation, too?
Please address the “need for further research” in the summary or another point of the publication.
Author Response
Dear Reviewer, thank you very much for all your comments and your concern for the improvement of our manuscript.
Following your suggestions we introduced many changes in the paper:
First, however, I would like to note that the current version does not contain line numbers, so it is difficult to point out all the corrections. Please take this into account in the next revision.
We used the template provided by MDPI in which all lines were numbered, but after we downloaded the version from the MDPI system there were no line number any more. We can not explain why it happened, probably it is some issue of formatting.
The manuscript still addresses an interesting topic. Overall, the manuscript has been revised mainly in form. There are no significant improvements in content. In my view, the manuscript needs to be revised at further points. I, therefore, ask the authors to address the following major points:
Introduction:
Overall, the introduction is short and not very informative. Only a few references are used. In particular, the definition of the research question is not comprehensible.
The interaction of HF, NO pathway, L-arginine derivative, and exercise capacity is not clearly described.
It is described that "little is known about variations in plasma ..." but no further details are given. What is already known about this?
Therefore, the aim of the study is not fully clear. Why is it essential to evaluate the L-arginine derivative changes after submaximal exercise?
We rewrote and reorganized the introduction following the above suggestions.
Methods:
The methods of the study are well-described, detailed, and comprehensible.
Please correct under point 2.3 "Statistical methods": Differences between Groups... Were two groups formed or do you mean "Within Group"?!
We corrected the part with statistical methods description. We clarified the definition of an end point.
Results:
There are important points to be revised in this part of the paper:
I cannot understand table 3. What are the reasons for choosing these parameters? Why is once the POST value was taken, once the PRE value, and once the "difference"? I don't see a pattern here. What about the other values? Why are many values missing in the table? The selection seems a bit random.
We analyzed the correlation between all measured L-arginine derivatives before exercise, after exercise, as well as the values of differences (value after exercise minus value before exercise) and the parameters of exercise capacity. Only the correlations presented in Table 3 were statistically significant (as it was already mentioned in the Table 3 legend). In the revised version we additionally described this in the text body in the results section. Choosing only statistically significant parameters (only 14 out of 63 analyzed correlations) were done on purpose due to simplify result presentation (instead of presenting huge, non-informative table with plenty of statistically non significant results). Moreover, in the table 3 we changed the sign “-“ for word “minus”
The risk of endpoint occurrence - how was this measured? Please explain a little more about "statistical methods". Are there different endpoints measured? In parallel to the correlations in table 3: what about the results of other arginine derivatives?
We haven’t analyzed the risk of end point occurrence at all but we tested whether arginine derivatives are predictors of end point occurrence. It is described in the method section (we used univariate Cox regression analysis). There is a statement in the results section that except from post-CPET SDMA concentration none of the other L-arginine derivatives had any impact on patients’ prognosis.
In total, the second part of the results section is not well structured. Please revise.
We revised result section in more structural way
Discussion:
The first half of the discussion refers to the basics of the introduction and does not include the actual results. Facts are mentioned here that are missing in the introduction. I, therefore, recommend that the introduction be supplemented with the contents of the discussion.
We rewrote and reorganized the introduction and the discussion.
The discussion of the results is well done. About the results section, is it also possible to make statements about a norm collective?
Concerning the collective norms: we haven’t found any information about standardized data providing norms for arginine derivatives in response to exercise. Results of experimental studies presenting such kind of data were mentioned in the discussion.
Please also address a possible clinical implication in the discussion.
Please address the “need for further research” in the summary or another point of the publication.
We added information about a need for further research in context of clinical implications.
In your opinion, is the lack of a control group a limitation, too?
We added information about lack of control group as a limitation of the study.
Round 2
Reviewer 3 Report
All comments have been addressed